# New Automated Method for Lung Functional Volumes Delineation with Lung Perfusion PET/CT Imaging

**DOI:** 10.3390/cancers15072166

**Published:** 2023-04-06

**Authors:** Fanny Pinot, David Bourhis, Vincent Bourbonne, Romain Floch, Maelle Mauguen, Frédérique Blanc-Béguin, Ulrike Schick, Mohamed Hamya, Ronan Abgral, Grégoire Le Gal, Pierre-Yves Salaün, François Lucia, Pierre-Yves Le Roux

**Affiliations:** 1Service de Médecine Nucléaire, CHRU de Brest, 29200 Brest, France; 2GETBO, INSERM, UMR1304, Université de Bretagne Occidentale, 29200 Brest, France; 3Radiation Oncology Department, University Hospital, 29200 Brest, France; 4LaTIM, INSERM, UMR 1101, University of Brest, 29200 Brest, France; 5Centre d’Investigation Clinique, CHRU de Brest, 29200 Brest, France

**Keywords:** 68Ga-MAA-lung perfusion PET/CT, lung radiotherapy, functional volumes, lung perfusion PET/CT

## Abstract

**Simple Summary:**

Lung perfusion PET/CT is an emerging imaging modality for the assessment of regional lung function, especially to optimise radiotherapy planning. A key step of lung functional avoidance radiotherapy is the delineation of lung functional volumes. We developed an automated relative to whole lung function segmentation method to delineate lung functional volumes. As compared to the commonly used relative to the maximal pixel value threshold method, this new approach allows for reproducible contouring, is relatively unaffected by the presence of hot spots, provides reliable and consistent functional volumes, and is clinically meaningful for clinicians.

**Abstract:**

Background: Gallium-68 lung perfusion PET/CT is an emerging imaging modality for the assessment of regional lung function, especially to optimise radiotherapy (RT) planning. A key step of lung functional avoidance RT is the delineation of lung functional volumes (LFVs) to be integrated into radiation plans. However, there is currently no consistent and reproducible delineation method for LFVs. The aim of this study was to develop and evaluate an automated delineation threshold method based on total lung function for LFVs delineation with Gallium-68 MAA lung PET/CT imaging. Material and Method: Patients prospectively enrolled in the PEGASUS trial—a pilot study assessing the feasibility of lung functional avoidance using perfusion PET/CT imaging for lung stereotactic body radiotherapy (SBRT) of primary or secondary lesion—were analysed. Patients underwent lung perfusion MAA-68Ga PET/CT imaging and pulmonary function tests (PFTs) as part of pre-treatment evaluation. LFVs were delineated using two methods: the commonly used relative to the maximal pixel value threshold method (pmax threshold method, X%pmax volumes) and a new approach based on a relative to whole lung function threshold method (WLF threshold method, FVX% volumes) using a dedicated iterative algorithm. For both methods, LFVs were expressed in terms of % of the anatomical lung volume (AV) and of % of the total lung activity. Functional volumes were compared for patients with normal PFTs and pre-existing airway disease. Results: 60 patients were analysed. Among the 48 patients who had PFTs, 31 (65%) had pre-existing lung disease. The pmax and WLF threshold methods clearly provided different functional volumes with a wide range of relative lung function for a given pmax volume, and conversely, a wide range of corresponding pmax values for a given WLF volume. The WLF threshold method provided more reliable and consistent volumes with much lower dispersion of LFVs as compared to the pmax method, especially in patients with normal PFTs. Conclusions: We developed a relative to whole lung function threshold segmentation method to delineate lung functional volumes on perfusion PET/CT imaging. The automated algorithm allows for reproducible contouring. This new approach, relatively unaffected by the presence of hot spots, provides reliable and consistent functional volumes, and is clinically meaningful for clinicians.

## 1. Introduction 

Gallium-68 lung perfusion PET/CT is an emerging imaging modality for the assessment of regional lung function [1]. Similar to conventional lung perfusion scintigraphy, the test assesses the regional distribution of pulmonary perfusion through the embolisation of macroaggregated albumin (MAA) in the pulmonary capillaries [2]. However, MAA particles are labelled with Gallium-68, a B+ emitter, instead of Technetium-99m. PET technology is intrinsically superior to SPECT with especially higher resolution and lower spatial resolution, thus improving the accuracy of regional lung function imaging [3,4]. Lung PET/CT imaging has already shown promising results in various clinical settings such as pulmonary embolism diagnosis [5,6], lung function assessment prior to lung surgery [7,8,9], and radiotherapy (RT) planning [8,9]. Functional lung avoidance RT is an emerging concept in the treatment of primary or secondary lung disease. The technique aims at personalising RT planning to individual’s pulmonary functional distribution, by prioritising delivery of higher dose in non-functional pulmonary regions while sparing functional areas [10,11].

A fundamental step of functional lung avoidance RT is the delineation of functional lung volumes to be integrated as organ at risk into RT planification. However, the definition of functional lung is not consistent throughout publications [10]. Some have proposed a manual contouring based on a visual evaluation of regional lung function [12]. However, this method is poorly reproducible and time-consuming, preventing its use in daily clinical practice. Most proposed automated contouring methods are based on the use of a fixed intensity threshold calculated as a percentage of the maximum pixel value within the lungs [10]. However, there is a very wide variety of thresholds used in the literature, ranging from 20% to 90% of the maximal value, questioning the robustness and reproducibility of the method [10]. Because a fixed threshold for all patients provides an inconsistent large range of functional volumes, a visually adapted semi-automatic method has also been proposed [13]. However, the visual adaptation of the threshold from patient to patient also prevents reproducibility.

Accordingly, in order to use lung perfusion imaging for functional lung avoidance RT, developing and validating a reliable and reproducible delineation method of lung functional volumes, less sensitive to the presence of hot spots, is required. An optimal delineation method should be fully automated or, at least, should not imply a visual adaptation that would limit reproducibility between physicians and centres. It should be consistent and not sensitive to the presence of hot spots. It should also be clinically meaningful, especially for radiation oncologists or chest physicians.

In that respect, we hypothesised that a delineation threshold method using as a reference for threshold determination the total activity within the lungs may be a consistent and reproducible way of delineating regional lung volumes. 

The aim of this study was to develop and evaluate an automated delineation threshold method based on total lung function for lung functional volumes determination with Gallium-68 MAA lung PET/CT imaging.

## 2. Methods 

### 2.1. Patients

The eligible population consisted of patients prospectively enrolled in the PEGASUS trial. The study was approved by the Nord Ouest IV Ethics Committee (ID RCB: 2021-002224-20) and registered in ClinicalTrial.gov registry (NCT04942275). Written informed consent was obtained from all participants. 

The PEGASUS trial is a pilot study assessing the feasibility to spare functional lung areas by integration of functional mapping guided by Gallium68-MAA perfusion PET/CT in lung stereotactic body radiotherapy (SBRT) planification [14]. Briefly, all patients were referred to the radiotherapy department of the University Hospital of Brest, France, for SBRT of a primary or secondary pulmonary lesion. As part of pre-treatment evaluation, patients underwent a lung perfusion MAA-68Ga PET/CT scan and pulmonary function tests (PFTs).

### 2.2. 68Ga-PET/CT Perfusion Protocol 

All patients underwent a lung perfusion PET/CT scan on a digital Biograph vision 600 PET/CT (Siemens Healthineers, Knoxville, TN, United States), immediately after intravenous administration of approximately 50 MBq of 68Ga-MAA [2,15].

Each acquisition consisted of, firstly, a low dose CT (120 kV, 10 mAs, pitch 1, 2 mm slices, FOV 780 mm) followed by a PET acquisition (motion mode 1.9 mm·s^−1^, total acquisition time about 5 min).

The OSEM 3D algorithm with time of flight (ToF) and point spread function (PSF) correction (TrueX + TOF) was used for reconstruction. PET acquisitions were corrected for random coincidence, scatter, deadtime, normalisation, isotope decay and attenuation with CT-data (transaxial recontruction 220 × 220, 4 iterations, 5 subsets, 4 mm gaussian post-filtering). No special instruction on breathing technique was given.

### 2.3. Lung Functional Volumes Delineation

All segmentations were performed on MIM image analysis software (MIM 7.3.3; MIMSoftware). 

First, an automatic contouring of the whole lung anatomical volume (AV) was performed on CT images based on Hounsfield unit values. AV contours were visually reviewed and modified if needed.

Lung functional volumes were then delineated within the AV using two methods: the commonly used approach based on a relative to the maximum pixel value threshold method, and a new approach based on a relative to whole lung function threshold method.

The relative to maximum pixel value threshold segmentation (pmax threshold method) was a classic relative threshold method. The segmented volume was defined inside the AV so that the selected pixels values had to be higher than the threshold. This threshold corresponded to a relative value of the maximum intensity within the AV. In this study, the chosen threshold values were 10, 20, 30, 40, 50 and 60% of the maximum intensity (X%pmax). Results were expressed in terms of volume (% of the AV) and counts (% of the total counts within the AV).

A new algorithm was developed for the relative to whole lung function threshold segmentation (WLF threshold method). The algorithm consisted in performing a large amount of pmax segmentations iteratively (from 5 to 95%, with 0.1% steps). For all segmented volumes, the relative functional value (FV) was calculated as follows: FV = total integrated counts within the segmented volume/total integrated counts within the AV. For example, a FV10% volume corresponded to the minimal volume containing 10% of the total counts within the AV. Thus, the following contours from 10 to 90% of total lung function were determined: FV10%, FV20%, FV30%, FV40%, FV50%, FV60%, FV70%, FV80%, and FV90%. Results were expressed in terms of volume (% of the AV). The corresponding pmax threshold was also reported.

### 2.4. Pulmonary Function Tests (PFTs)

As part of the PEGASUS study, patients underwent PFTs including spirometry, plethysmography, and DLCO. Patients were categorised as having lung disease if they presented with chronic obstructive pulmonary disease (COPD) with FEV1/FVC < 0.7 according to GOLD classification, or a diffusion disorder with DLCO < 60% [16].

### 2.5. Statistics 

Descriptive statistics were performed using Excel. A parametric t-student test was used to compare the functional volume values between the population with normal versus pathological PFTs with the XLSTAT software. The null hypothesis was rejected when *p* was less than 0.05.

## 3. Results

### 3.1. Study Population 

Sixty patients were included in the prospective clinical PEGASUS trial between July 2021 and January 2022 [14]. All were referred for SBRT for a primary or secondary lung lesion. Patients’ characteristics are presented in Table 1. Median age was 69 years (IQ 63–72.3) and 29 (48%) were female. In total, 43 (72%) patients were former or current smokers. Additionally, 19 (32%) had previous thoracic radiotherapy and 14 (23%) had prior thoracic surgery. 

Twelve (20%) patients did not perform PFTs. Out of the 48 patients who underwent PFTs, 31 (65%) had a pre-existing airway disease (FEV1/FVC < 0.7 or DLCO < 60%). 

### 3.2. Pmax Threshold Method 

The distribution of functional volumes computed with the pmax threshold method, expressed as a percentage of the AV, in the 60 patients is shown in Table 2 and Figure 1. The median with inter-quartile (IQ) volumes were 93.8% (88.2–96.4) for 10%pmax, 78.2% (65.3–85.8) for 20%pmax, 54.2% (40.8–67.7) for 30%pmax, 34.1% (19.6–47.3) for 40%pmax, 18.3% (7.1–28.5) for 50%pmax, and 6.6% (2.2–13.3) for 60%pmax, respectively. The correspondence with the % of WL activity is displayed in Table 2.

### 3.3. Relative to Whole Lung Function Threshold Method

The distribution of functional volumes computed with the relative to WLF threshold method, expressed as a percentage of the AV, in the 60 patients, is shown in Table 2 and Figure 1. The median (IQ) volumes were 5.0% (4.4–5.5) for FV10%; 10.9% (9.7–11.8) for FV20%; 17.6% (16.1–18.8) for FV30%; 24.4% (23.0–26.3) for FV40%; 32.6% (30.7–34.2) for FV50%; 41.4% (39.0–43.1) for FV60%; 51.9% (48.3–53.3) for FV70%; 62.8% (60.0–64.5) for FV80%; and 76.1% (73.7–78.2) for FV90%, respectively. The correspondence with the pmax threshold values is presented in Table 2. Figure 2 shows examples of heterogeneous distribution of FV50% computed with the WLF threshold method.

### 3.4. Comparison of Functional Volumes in Patients with or without Lung Disease 

The distribution of functional volumes computed with the pmax threshold method in patients with (n = 31 patients) and without (n = 17 patients) lung disease is displayed in Figure 3. Similarly, the distribution of functional volumes using the WLF method is displayed in Figure 4. No statistical difference was found between patients with and without lung disease.

## 4. Discussion

In this study, we aimed to develop an automated threshold method for lung functional volumes delineation with perfusion 68Ga-PET/CT imaging, using WLF as a reference to determine the threshold values.

Lung functional mapping is a key step of functional lung avoidance RT planning, especially with the advent of SBRT [11]. However, there is currently no consensual method for lung functional volumes delineation. Some have proposed manual or visually adapted semi-automatic delineation methods [12,13]. However, such methods prevent reproducibility between physicians and centres. Most of automated methods are based on the use of a fixed threshold value, determined as a percentage of the maximum value within the lungs [10]. The inherent limitation of this method is that it only relies on the value in a single pixel, which is subject to great variability (image noise, presence of hot spots, etc.). Furthermore, the highest pixel value within the lungs has no particular significance nor clinical correlation with lung function. This likely explains why there is such a wide range within the published literature of the thresholds used for lung functional volumes delineation [10]. Similarly, the pmax threshold method provided a wide range of lung functional volumes in our series. This wide range of functional volumes for a given pmax threshold is particularly questionable in patients with normal PFTs. For example, the extrema of the 50%pmax volumes was 7.4% to 41.8% of the AV.

We aimed to propose a more relevant and reliable reference to compute the threshold values for lung function delineation. In this respect, the total activity within the lungs, which can be regarded as total lung function, appears as a more accurate reference. Another advantage of the method is that it is clinically meaningful. For instance, the FV50% is the minimal volume within the lungs which contains 50% of total activity, while the 50%pmax volume has no clinical significance.

To the best of our knowledge, this approach has never been proposed so far. The delineation was performed using a dedicated algorithm which consisted in performing a large amount of pmax segmentations iteratively, with calculation of the corresponding relative activity within the segmented volumes as compared with the total counts in the AV, until the various threshold values were reached.

One clear result of our study is that the pmax and WLF threshold methods definitely provide different lung functional volumes. For example, the 25 and 75% interquartile of the percentage of total activity within the 50%pmax volume were 14.0 and 45.3%, respectively. Conversely, the 25 and 75% interquartile of the corresponding pmax values of the FV50% volume was 33.8 and 48.0, respectively. Secondly, in contrast to the pmax threshold method which provided a wide distribution in the size of functional volumes, the relative to WLF threshold provided much more consistent and reliable results between patients with lower dispersion of functional volumes. Thus, the median (IQ) of the FV50% in patients without lung disease was 32.6% (30.7–34.2) and the range of 30.9–36.7% of the AV, respectively. However, while the size of the FV was close between patients, the spatial distribution was variable from one patient to another, as illustrated on Figure 2.

We failed to demonstrate a relation between FV and PFTs parameters using both delineation methods. There was no significant difference in the various FV in patients with normal lung function and abnormal PFTs. This can potentially be explained by our small sample size of patients. However, there was a trend to observe lower functional volumes in patients with lung diseases as compared with patients with normal PFTs (See Figure 3 and Figure 4).

Our study was performed on lung perfusion PET/CT imaging, without prior administration of Ga-68 labelled carbon nanoparticles for ventilation imaging. In this setting, it is likely that the relative to WLF method would be even more relevant than the relative to pmax threshold method, as this method is much more sensitive to the presence of hot spots. Indeed, hot spots are localised accumulation of very high radioactivity. On perfusion imaging, hot spots can be caused by MAA clumping due to faulty injection technique. However, they are most commonly observed on ventilation imaging with aerosolised particles, especially in patients with impaired lung function, due to the accumulation of particles in the proximal bronchial airways. However, when the ventilation is performed before the perfusion, hot spots from the ventilation can still be observed on perfusion images. This very high artefactual accumulation of radioactivity makes the pmax method irrelevant. Furthermore, the relative to WLF threshold method could be easily implemented for lung SPECT/CT imaging. Finally, we included patients planned to be treated with SBRT for stage I lung cancers or metastases from other cancers. Patients with locally advanced lung cancer treated with conventional radiotherapy are more likely to have impaired and heterogeneous lung function. Again, the WLF method would be even more relevant in this setting.

A limitation of the study is that we did not compare LFV to a reference standard. Indeed, lung function is not binary. In a given territory, lung function is not, either perfectly functional or non-functional at all. Between the two extremes, there are various degree of function in the lung territories, especially in our population including 55% of patients with pre-existing lung disease. Especially in a RT planning perspective, it was not possible to identify one single LFV that would have been considered a gold standard. In contrast, we aimed to propose a method that would consistently delineate lung volumes with varying degrees of lung function. Another limitation is the small cohort of 60 patients, including 48 patients with pulmonary function tests. It would be of interest to confirm these results in a larger independent cohort of patients.

Based on the results of this study, some functional volumes appear to be of particular interest. The FV50% may be seen as highly functional lung, as 50% of total lung function is contained within this volume corresponding to approximately 30% of the AV. In contrast, the lung volume outside the FV90% may be seen as poorly functional lung with only 10% of WLF contained in this volume corresponding to 25% of the AV. An intermediate lung functional volume is the FV70%, accounting for approximately 50% of the AV.

## 5. Conclusions

We developed a relative to whole lung function threshold segmentation method to delineate lung functional volumes on perfusion PET/CT imaging. The automated algorithm allows for reproducible contouring. This new approach, relatively unaffected by the presence of hot spots, provides reliable and consistent functional volumes, and is clinically meaningful for clinicians.

## Figures and Tables

**Figure 1 cancers-15-02166-f001:**
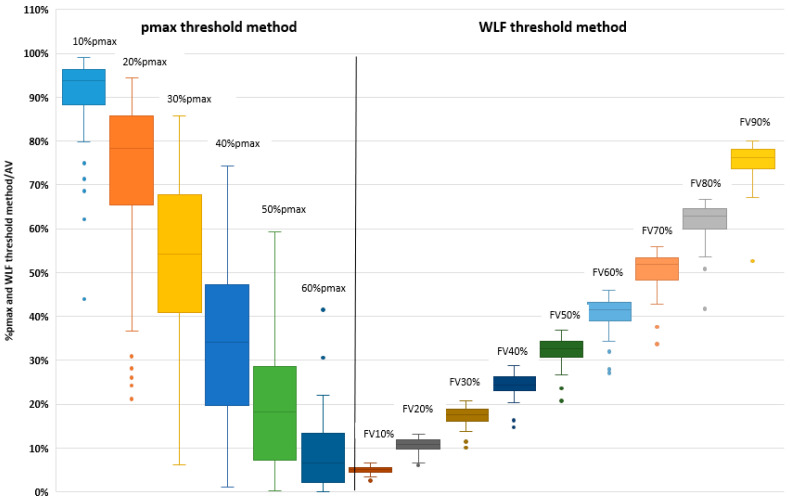
Functional volumes (FV) relative to AV using pmax threshold method (left) and WLF threshold method (right).

**Figure 2 cancers-15-02166-f002:**
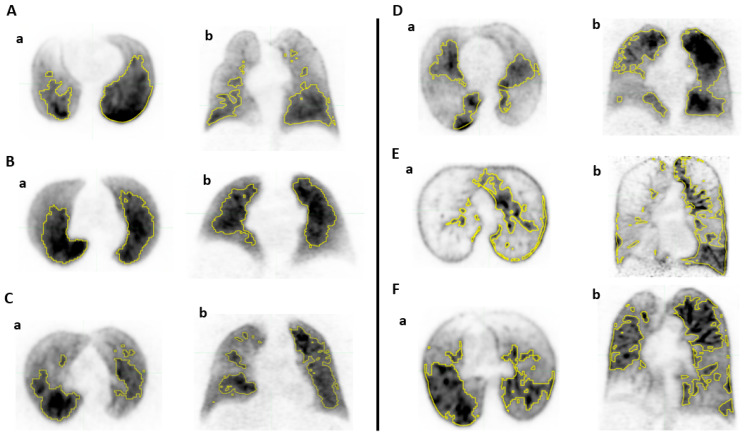
Illustration of perfusion volume FV50 using WLF threshold method in axial (**a**) and coronal section (**b**) in patients with normal (**A**–**C**) and abnormal (**D**–**F**) PFTs.

**Figure 3 cancers-15-02166-f003:**
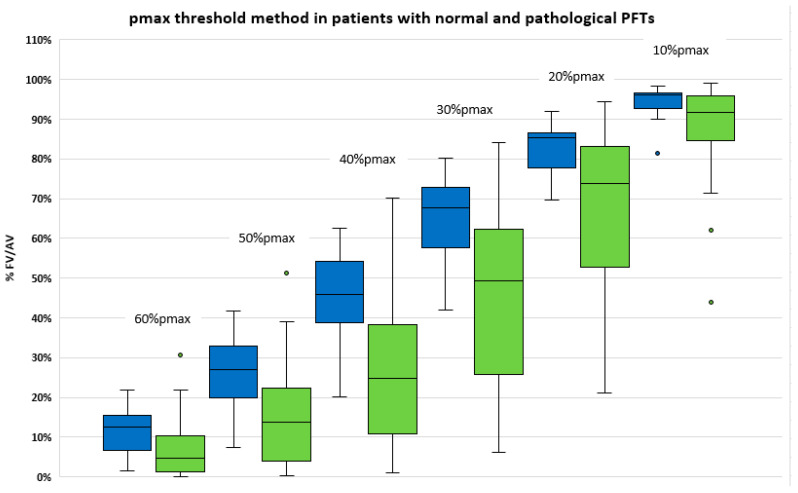
Functional volumes (% AV volume) using pmax threshold method in patient with normal PFT (left, blue) and with pre-existing airway disease (right, green) (FEV1/FVC < 0.7 or DLCO < 60%).

**Figure 4 cancers-15-02166-f004:**
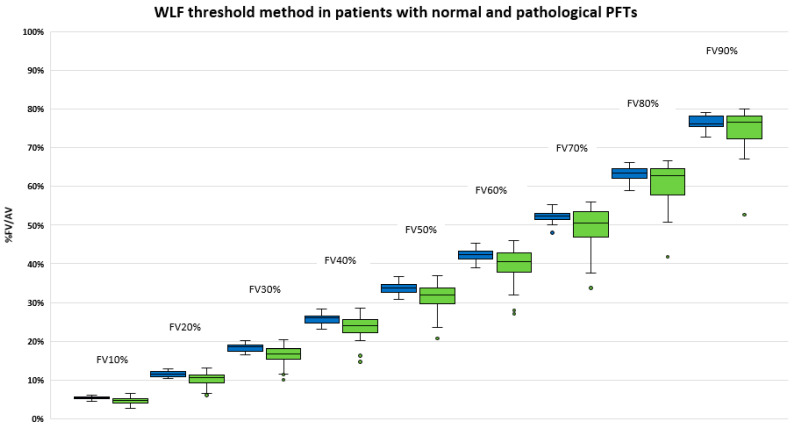
Functional volumes (% AV volume) using WLF threshold method in patient with normal PFT (left, blue) and with pre-existing airway disease (right, green) (FEV1/FVC < 0.7 or DLCO < 60%).

**Table 1 cancers-15-02166-t001:** Characteristics of the population.

Patients Characteristics		
General Characteristics		No. of Patients (%)
No. of patients		**60**
	Age (y)	69 (IQ 63.0–72.3)
	Men	31 (52%)
	Women	29 (48%)
Performans status		
	PS 0	27 (45%)
	PS 1	23 (39%)
	PS 2	8 (13%)
	Unknown	2 (3%)
Tobacco smoking		
	Unknown	2 (3%)
	Never-smoker	15 (25%)
	Former or current smoker	43 (72%)
Previous thoracic treatment		33 (55%)
	Radiotherapy	19 (32%)
	Lung surgery	14 (23%)
PFTs		
	Not performed or not interpretable	12 (20%)
	Normal	15 (25%)
	Abnormal (FEV1/FVC < 0.7 or DLCO < 0.6)	33 (55%)

**Table 2 cancers-15-02166-t002:** Lung functional volumes expressed in % of the AV and in % of the total activity in the AV for the pmax threshold method. Lung functional volumes expressed in % of the AV and the corresponding pmax threshold for each FV% for the WLF threshold method.

**Pmax Threshold Method**		
	**Volume (% of the AV)** **Median (IQ)**	**Activity (% of Total Activity in the AV)** **Median (IQ)**
10%pmax	93.8 (88.2–96.4)	98.9 (97.4–99.4)
20%pmax	78.2 (65.3–85.8)	92.0 (86.5–95.2)
30%pmax	54.2 (40.8–67.7)	75.5 (62.1–84.0)
40%pmax	34.1 (19.6–47.3)	55.3 (34.0–66.9)
50%pmax	18.3 (7.1–28.5)	32.3 (14.0–45.3)
60%pmax	6.6 (2.2–13.3)	14.5 (5.6–24.3)
**WLF Threshold Method**		
	**Volume (% of the AV)** **Median (IQ)**	**Corresponding %pmax threshold** **Median (IQ)**
FV10%	5.0 (4.4–5.5)	63.0 (52.8–69.3)
FV20%	10.9 (9.7–11.8)	55.5 (46.0–62.3)
FV30%	17.6 (16.1–18.8)	51.0 (41.0–57.0)
FV40%	24.4 (23.0–26.3)	46.5 (37.0–52.0)
FV50%	32.6 (30.7–34.2)	42.0 (33.8–48.0)
FV60%	41.4 (39.0–43.1)	38.0 (30.8–43.3)
FV70%	51.9 (48.3–53.3)	33.0 (26.5–38.3)
FV80%	62.8 (60.0–64.5)	27.0 (22.8–33.0)
FV90%	76.1 (73.7–78.2)	21.0 (17.0–26.0)

## Data Availability

The original contributions presented in the study are included in the article, further inquiries can be directed to the corresponding authors.

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
