# Peer review of "New Automated Method for Lung Functional Volumes Delineation with Lung Perfusion PET/CT Imaging"

_cancers, 2023, doi:10.3390/cancers15072166_

Round 1

Reviewer 1 Report

Interesting new approach to objective the functional parts of the lung before SBRT, with less heterogeneity than what was used before.

A simple and bright idea !

Author Response

Dear Editor, Dear reviewers,

We would like to thank you for giving us the opportunity to resubmit a revised version of our manuscript “New Automated Method for Lung Functional Volumes Delineation with Lung Perfusion PET/CT Imaging.”

Please find below our point-by-point reply to the reviewers’ comments. We have taken these comments into account.

To make the reading easy, all the changes into the manuscript were made in “Red” font. We do hope that this revised version of our manuscript is now suitable for publication.

Looking forward to hearing from you,

Sincerely,

Dr Fanny Pinot

Prof Pierre-Yves LE ROUX

Reviewer 1 :

Interesting new approach to objective the functional parts of the lung before SBRT, with less heterogeneity than what was used before. A simple and bright idea!

We thank the reviewer for this positive feedback.

Reviewer 2 Report

In this interesting article the authors present a new method to delineate lung function volumes automatically delineated on Ga-68 PET-CT scan. The key message is that the threshold – either pmax or the new WLF – leads to different results. From this study it seems that the pmax method is clinically not very relevant as opposed to the WLF method. To me this seems plausible.

Before publication, the following aspects merit consideration.

1.       Please provide a patient table with the patient characteristics.

2.       Main limitation: I believe that the main limitation of this study is the small patient cohort. It seems to me that this cohort of 48 patients could serve as a pilot (as stated), but the result should be validated in an independent cohort of patients.

3.       These patients are stage I lung cancers or metastases from other cancers. About 65% of the patients (as stated by the authors) have impaired lung function because of pre-existing lung damages. It would be good if the authors could tie back their findings to locally advanced lung cancer stages, which are more likely to have lung diseases with clinically relevant lung function impairment prior to treatment.

4.       Figure 2. Could the authors please provide the respective radiation treatment plans for the patients shown here.

5.       The authors use the term pixel throughout the manuscript – should it not be voxel?

6.       Lines 212 – 217. I fully agree.

7.       Lines 243 – 244. What exactly is a hot spot? Please explain.

8.       Lines 247 – 248. What does this exactly mean? Does it mean that some anatomical regions like lower lobes are physiologically more relevant for gas exchange? Please explain.

9.       The limitation section of the discussion should be expanded.

Round 2

Reviewer 2 Report

I have nothing to add.